# A Study on The Carriers Compound Multi-Stage MBBR Biological Treatment Process for Domestic Sewage

**Miaojie Li [1], Yonghong Liu [2,\*], Xiaode Zhou [1], Ning Wang [2] and Bo Yuan [1]**

1   College of Geology and Environment, Xi'an University of Science and Technology, Xi'an 710054, China; leemj1028@163.com (M.L.); 1044176423@139.com (X.Z.); boyuan@xust.edu.cn (B.Y.)
2   College of Environmental and Chemical Engineering, Xi'an Polytechnic University, Xi'an 710699, China; ninaw2018@163.com
\*   Correspondence: 13572505762@139.com

**Abstract:** Biocarriers are a key factor in moving bed biofilm reactors (MBBR) for domestic wastewater treatment and studies targeting biocarriers can be explored in more depth. In this study, two different types of biocarriers, namely anaerobic microbial carriers (AMC) and porous biogels (PBG), were used to treat real domestic wastewater and acted on a lab-scale tertiary MBBR system. The effects of the start-up process, water quality degradation, secondary start-up, and organic loading rate (OLR) on MBBR performance at room temperature (19–24 °C) and the same filling ratio (40%) were investigated, as well as the calculation of sludge yield. The results showed that the AMC–PBG/MBBR biological treatment process could complete the start-up process quickly in a short time and the OLR was finally determined to be 1.5 kgCOD/(m$^3$·d). In this case, the system was able to operate stably and complete the secondary start-up relatively quickly, with 80% and 95% removal of COD and NH$_4^+$-N, respectively. The biofilm was characterized by scanning electron microscopy (SEM) and high-throughput sequencing which revealed the changes of microorganisms in the biofilm during operation, among which the dominant phyla were *Euryarchaeota* and *Proteobacteria*. Finally, the apparent production of process sludge was monitored and calculated to be 0.043 kgMLSS/kgCOD which is a significant reduction in sludge compared to the conventional activated sludge method. These conclusions provide valuable information for the full-scale treatment of domestic wastewater.

**Keywords:** domestic sewage; organic loading rates; novel carrier; sludge yield rate

## 1. Introduction

The main purpose of municipal wastewater treatment plants (WWTP) using conventional activated sludge (CAS) is to use the activity of microbial communities to neutralize ammonia and reduce nitrate. However, the effluents of conventional WWTPs still contain large amounts of micropollutants (MPs) which are continuously released into the aquatic environment and thus pose a serious threat for the aquatic environment [1–3]. Although deep treatment by physicochemical methods such as ozone oxidation or activated carbon treatment is extremely effective, there are problems of membrane contamination, high energy loss, high cost, and secondary contamination that limit the application of physicochemical methods, such as membrane contamination, high energy loss, high cost of drug addiction, and secondary contamination in the process. Although advanced treatment by physicochemical methods such as ozone oxidation or activated carbon treatment is more effective, biofilm-based treatment processes, such as moving bed biofilm reactor (MBBR) technology, may also be options to improve MP removal and they are heavily used in various wastewater treatment configurations [4].

The MBBRs are cost-effective continuous processes with a small footprint, easy installation, control and operation, and minimal sludge production (90% lower than traditional WWT) [5–8]. Compared to conventional activated sludge reactors, MBBR technology offers

compact treatment capacity without eliminating the potential of the treatment itself. It has been successfully applied in various areas of wastewater management, including municipal wastewater, dairy wastewater, pulp and paper wastewater, and seawater [9–11]. Some of the advantages that determine the appropriateness of MBBR technology in the industry are its simplicity of operation and design, lack of media clogging, high efficiency, and effluent quality [12]. In addition, the reactor is relatively tiny, resistant to shock loading, has a small footprint, does not require periodic backwashing, and does not suffer from sludge expansion [13–16]. In MBBR systems, the biomass consists of suspended sludge and biofilm grown on specially designed carriers, which are kept in constant motion in a continuously mixed bioreactor by aeration and hydrodynamic agitation by mechanical stirring or liquid jets [10,17,18].

In contrast to CAS, the carriers provide a surface area for the adherence of slower-growing bacteria, thus promoting the development of specific microbial communities and increasing the biotransformation of several MPs as it directly controls bacterial growth and reproduction during biofilm formation [18–20]. Therefore, the core component of the MBBR system is the biological carrier on which the bacteria adhere to form a biofilm structure, the properties of which play a crucial role in influencing the performance of biological nitrification [21,22]. The low efficiency of MBBR filled with conventional bio-carriers may be related to reduced surface hydrophilicity and biocompatibility, limited denitrification capacity, reduced mass transfer capacity, and prolonged biofilm growth initiation period [22,23]. Several researchers have evaluated the suitability of different kinds of materials as MBBR biocarriers. For example, novel carbon-based biocarriers, RK bioelement carriers, modified plastic biocarriers containing zinc nanoparticles, saddle-shaped Z-MBBR biocarriers, sponge biocarriers (SB), K3 polystyrene carriers, LEVAPOR carriers, and immobilized microbial granules [14,15,22,24–26].

Therefore, the main objectives of this study were to examine the ability of the packed composite multistage MBBR system to treat domestic wastewater, water quality analysis focusing on COD, $NH_4^+$-N and mixed liquor suspended solid (MLSS), observation of microorganisms attached to both carriers by microscopy and scanning electron microscopy, identification of microbial diversity attached to the biofilm carriers using high-throughput sequencing, calculation of sludge production during stable operation of the system, and comprehensive evaluation of the packed composite. The system was used to calculate the sludge yield during stable operation, to evaluate the operation effect of the composite MBBR system, and to provide theoretical reference for the wastewater treatment process.

## 2. Materials and Methods

### 2.1. Configuration

Domestic sewage and activated sludge were collected from a drainpipe at Xi'an Polytechnic University (Xi'an, China). The basic data of domestic sewage are listed in Table 1.

**Table 1.** The basic data of domestic sewage.

| Kind | pH | COD [mg/L] | $NH_4^+$-N [mg/L] | $BOD_5$ [mg/L] | Turbidity [NTU] | SS [mg/L] |
|---|---|---|---|---|---|---|
| Basic data | 7.5–8.0 | 100–470 | 18–67 | 120–250 | 50–220 | 100–400 |

Table 2 shows two types of carriers with different technical parameters used for the experiment. The AMC carrier is a new type of carrier of organic–inorganic composite material, such as polyvinyl alcohol (PVA), and has the advantages of low cost, rough surface, and easy microbial adhesion but it is hard. The PBG carrier is a porous flexible polyurethane urea sponge, similar in shape to a 10 mm cube, slightly swelling in water, and is a typical biomass carrier. The combination of the two carriers provides a more suitable habitat for different species of bacteria.

**Table 2.** The characteristics and working parameters of two carriers.

| Items | AMC | PBG |
|---|---|---|
| Photo of carriers |  |  |
| SEM of carriers |  |  |
| Shapes | Cylinder | Cube |
| Size | 6–8 ± 1 mm | 10 ± 1 mm |
| Bulk density | 400 kg/m$^3$ | 12.5 ± 0.75 kg/m$^3$ |
| Specific gravity | Slightly more than 1 | Close to 1 |
| Surface shape | Convex concave | Spongy |
| Interior structure | 5–10 μm porous | Porous |
| Porosity | / | 98% |
| Specific surface area | / | >4000 m$^2$/m$^3$ |
| Application | AMC Reactor | PBG Reactor |
| Filling ratio | 40% | 40% |

*2.2. Experimental Device and Start-Up*

2.2.1. Lab-Scale Aerobic Reactors System

The experimental system mainly consists of an AMC reactor (25 L), a two-stage PBG reactor (25 L), and a sedimentation tank (10 L), as shown in Figure 1. Each reactor was made of plexiglass of 25 cm in length, 25 cm in width, and 45 cm in height. The wastewater entered the reactor from the bottom up and was thoroughly mixed with the carrier under the action of airflow to ensure the uniformity of the carrier distribution in the liquid and to provide an aerobic environment for bacteria [27]. The inlet water flow was controlled by a variable-speed peristaltic pump. The aeration unit was arranged at the bottom of the AMC or PBG and activated sludge reactor and was supplied by an oxygen pump.

2.2.2. Reactors Start-Up

AMC and PBG reactors were established and inoculated with flocculated aerobic sludge, respectively. The concentration of suspended solids (MLSS) in the mixture of inoculated sludge in all three reactors was around 5563 mg/L. Based on the laboratory experimental data and references, AMC and PBG reactors were introduced with a filling rate of 40% and the total amount of carriers in each reactor was 5 L. Then, 11.5 L of activated sludge and 10 L of simulated wastewater with a carbon-to-nitrogen ratio of 40:1 were added to the AMC reactor. Equal amounts of activated sludge and equal volumes of PBG carriers were added to the first stage PBG reactor and the second stage PBG reactor and mixed for 5 days. The reactors were started at organic loading rates (OLR) of 0.1 kgCOD/(m$^3$·d) to 0.5 kgCOD/(m$^3$·d) and continuously injected with actual domestic wastewater [14,28,29].

During the experimental start-up phase, the reactor was conducted at room temperature (19–24 °C).

**Figure 1.** Flow diagram of the AMC–PBG/MBBR biological process.

The influent pH ranged from 6.5 to 8.0 and DO ranged from 3.5 to 6.0 mg/L. The AMC–PBG/MBBR process was started with an OLR of 0.5 kgCOD/m$^3$ d, an HRT of 48.6 h, and an average temperature of 18.5 °C. The average daily liquid intake was 42 L and the average mass concentrations of COD and $NH_4^+$-N were 186.7 mg/L and 38.01 mg/L, respectively. After 5 days of operation, the removal rates of COD and $NH_4^+$-N exceeded 80% and 98%, respectively. The efficiency of the AMC–PBG/MBBR reactor was stable, which means that the start-up of the AMC–PBG/MBBR reactor had been completed.

*2.3. Analytical Methods*

COD and $NH_4^+$-N concentrations were measured daily with a three-parameter water quality rapid tester (5B-3C, Lianhua environmental protection technology co. Ltd., Lanzhou, China). DO was monitored daily in the field using portable equipment. TN was measured using a WZS-185 turbidimeter (INESA Scientific Instrument Co., Ltd., Shanghai, China). The MLSS and SS of the wastewater and carriers were measured by the weighing method every three days. The activity of microorganisms attached to the biological carriers or floating in the reaction tank was observed daily by biomicroscopy (N-180 M, NOVEL company, Ningbo, China).

Several carriers were randomly selected as samples at different stages of the run. Carrier samples were washed gently with 50 mM phosphate buffer (pH 7.0), fixed with glutaraldehyde in phosphate solution in buffer solution (2.5% *w/v*, pH 7.0), and left for 12 h. The fixed carrier samples were dehydrated with ethanol and dried in a constant temperature drying oven at 35 °C [27]. The morphology of carriers was obtained with a field emission scanning electronic microscope (SEM) (Quanta 600 FEG, FEI Company, Hillsboro, OR, USA). The extracted sludge suspension is sent to Sangong Biotechnology (Shanghai, China) Co., Ltd., for further high-throughput sequencing analysis.

**3. Results and Discussion**

*3.1. Finding the Best OLR of the System*

During start-up (Figure 2), the system was operated at a low OLR (0.5 kgCOD/(m$^3$·d)) to ensure microbial acclimation and growth [30]. Each load stage was operated for about

5 days and the removal rates of COD and $NH_4^+$-N were tested. If they were stabilized above 80%, upgrading the compound was considered until it reached 2.0 kgCOD/(m$^3$·d).

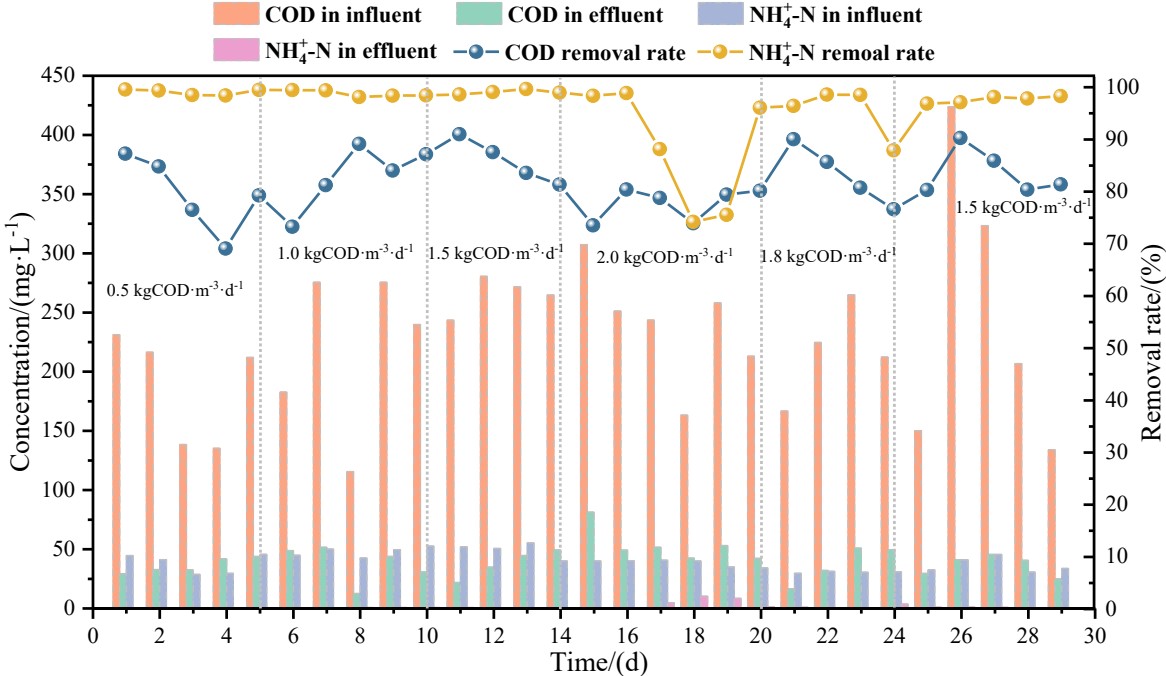

**Figure 2.** Treatment efficiency of the system during start-up.

As can be seen from Figure 2, on days 4–14 the COD removal efficiency of the process basically showed an increasing trend and the removal efficiency was stable above 80% on days 7–14, with the highest removal efficiency reaching 90%. When the OLR was 2.0 kgCOD/(m$^3$·d), the COD removal rate fluctuated and the removal rate was basically below 80%, with an average removal rate of 77%. On days 21–24, the OLR decreased to 1.8 kgCOD/(m$^3$·d) and the removal rate increased, reaching 90.1% on day 21, then gradually decreased to 76.6%. When the OLR was finally adjusted to 1.5 kgCOD/(m$^3$·d), the COD removal rate began to rise, reaching a maximum of 90.3% and finally stabilized above 80%.

In addition, increasing the OLR had little effect on the removal of $NH_4^+$-N, with all removal rates above 96%. Only when the OLR was 2.0 kgCOD/(m$^3$·d) and 1.8 kgCOD/(m$^3$·d), respectively, did the removal rate of $NH_4^+$-N by the process decrease significantly (75% and 87%, respectively). When partial denitrification proceeded, $NH_4^+$-N decreased, which indicated that a certain amount of accumulated $NO_2^-$-N and incoming $NH_4^+$-N were removed by anaerobic bacteria during the $NO_2^-$-N accumulation period. After the depletion of $NO_3^-$-N, the accumulated $NO_2^-$-N acts as an electron acceptor and removes $NH_4^+$-N through the anaerobic process [31]. So, it was speculated that after upgrading the OLR, a large amount of domestic wastewater enters the reactor and the $NO_3^-$-N produced by anaerobic bacteria cannot be reduced to $NO_2^-$-N by endogenous denitrification, and subsequently cannot be removed together with $NH_4^+$-N, which leads to an increase in the $NH_4^+$-N removal rate. To maintain the stability of the process, the OLR was gradually reduced to 1.5 kgCOD/(m$^3$·d) and operated stably for 20 days.

The results showed that the COD removal rate increased within 1–2 days after each increase of the primary OLR until it was raised to 2.0 and 1.8 kgCOD/(m$^3$·d), where the COD and $NH_4^+$-N removal rates decreased significantly. The optimal OLR of the AMC–PBG/MBBR biological treatment process for treating the domestic wastewater of the school was 1.5 kgCOD/(m$^3$·d). Under this OLR operation condition, the system achieved 80% and 96% removal of COD and $NH_4^+$-N, respectively; meanwhile, the removal

rate of BOD$_5$ reached more than 94%, indicating that the AMC–PGB/MBBR process has a stable and high removal efficiency in the load-raising stage.

### 3.2. Secondary Start-Up and Stable Operation

To investigate the stability of the process operation under the above conditions, the process was settled for 3 and 6 days at the optimum OLR (1.5 kgCOD/(m$^3$·d)), respectively. At this time, the HRT was 16.3 h. The effect of the AMC–PBG/MBBR biological treatment process on the actual domestic wastewater during operation is shown in Figure 3.

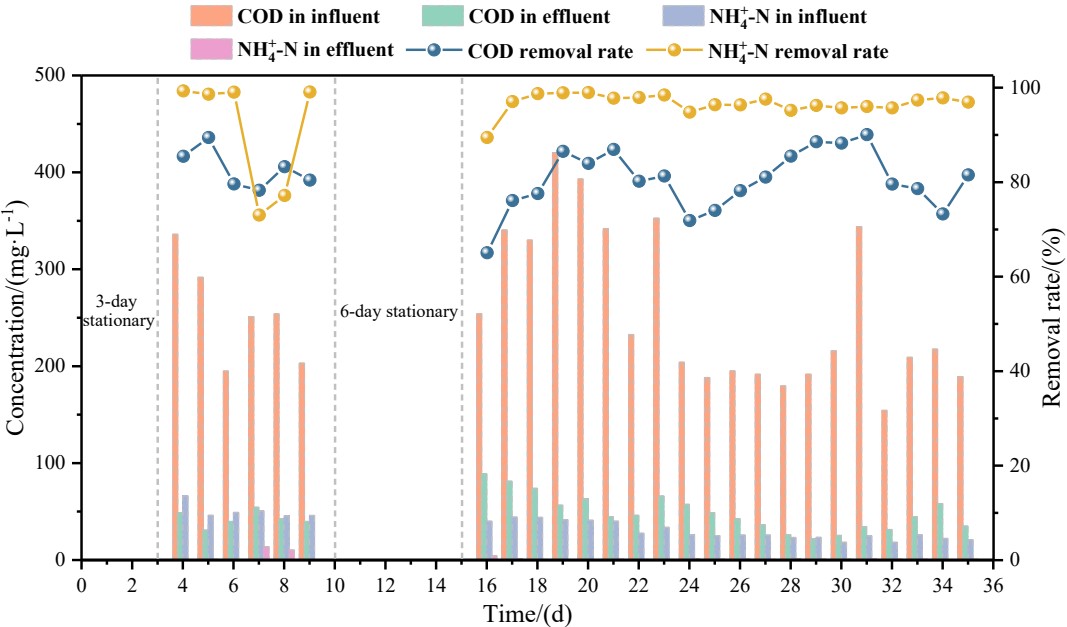

**Figure 3.** Treatment efficiency of the system during the second start-up.

Based on Figure 3, days 1 to 6 showed the results of the operation after the second start-up. After 3 days of settling, the removal efficiency of COD and NH$_4^+$-N reached 85.5% and 99.3%, respectively, on the first day. Subsequently, the COD removal efficiency increased and then decreased due to the decrease in influent concentration, but the COD concentration was still guaranteed to be below 50 mg/L with a minimum of 30.78 mg/L. In addition, the NH$_4^+$-N removal rate also showed a trend of increasing and then decreasing due to the decrease in DO [32]. After the adjustment of DO, the NH$_4^+$-N removal rate started to increase.

The process was then left to stand for 6 days. The results of the second secondary start-up and operation were shown in Figure 3 and lasted from 7 to 26 days. It can be seen that the COD removal rate was only 65.1% on the first day and reached 76.1% on the second day. On the 6th day, the removal rate reached 86.9% and the effluent concentration decreased to 44.6 mg/L. During the stabilization stage, the removal rate still showed a trend of decreasing and then increasing because the influent COD concentration was too low. The average influent COD from day 15 to day 21 was only 194 mg/L, but the average effluent COD concentration was 35.5 mg/L and the minimum effluent concentration was 21.9 mg/L, with a removal efficiency of 90%, indicating that the process had a significant and stable effect on COD removal. Meanwhile, the removal rate of NH$_4^+$-N was only 89.5% on the first day and also increased significantly to 97.0% on the second day. During steady operation, the removal efficiency remained above 95.5% and reached 99.0% with a final effluent concentration of 0.64 mg/L. The reason could be that there was sufficient HRT (average 5.4 h per reactor) and dissolved oxygen (3–4 mg/L) in each reaction tank [33], which allowed the process to have a strong nitrification capacity. The effluent BOD$_5$, SS, and TN were reduced to below 20 mg/L, 20–40 mg/L, and 10 NTU, respectively. The effluent water quality

all met the National *Emission Standards for Pollutants from Urban Sewage Treatment Plants* (GB 18918-2002), namely the first-level standard (SS reached the second-level standard), indicating that the process has a strong secondary start-up capability, which also confirms Shore's study [16]. It showed that the AMC–PBG/MBBR biological treatment process has outstanding technical advantages and engineering value.

*3.3. Characterization of Biocarriers*

In order to analyze the microbial changes of the two types of carriers during operation and the mechanism of their effects on wastewater treatment, AMC and PBG were sampled and characterized in different operation stages.

After one month of operation, the AMC carrier changed from the initial yellow color to brown color and at the end of the operation (80 days) the mature AMC turned dark brown, as shown in Figure 4. Meanwhile, the rough surface and porous structure of AMC caused the activated sludge to adhere to the surface. The change in the apparent color of AMC indicated that the number of microorganisms attached to AMC increased, leading to the deepening of the color.

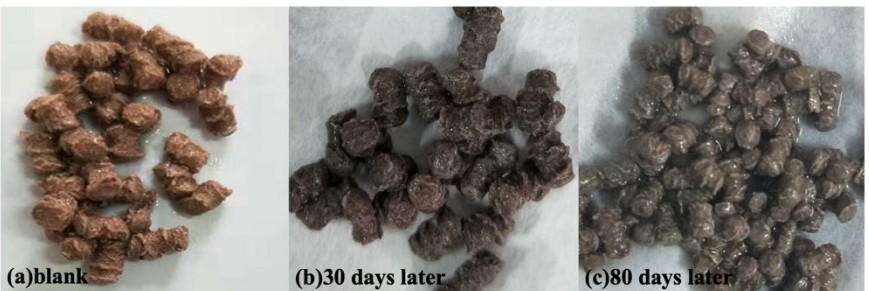

**Figure 4.** Change of the AMC carriers.

The changes in the appearance of PBG after one month of operation are shown in Figure 5a–c. Due to the microporous structure of PBG carriers and their unique hydrophilic gel properties, the membrane size increased by about 10 mm after a period of incubation due to water absorption and swelling. At the end of the operation (80 d), the increase in activated sludge caused the color of PBG to change from white to yellow and finally to dark brown.

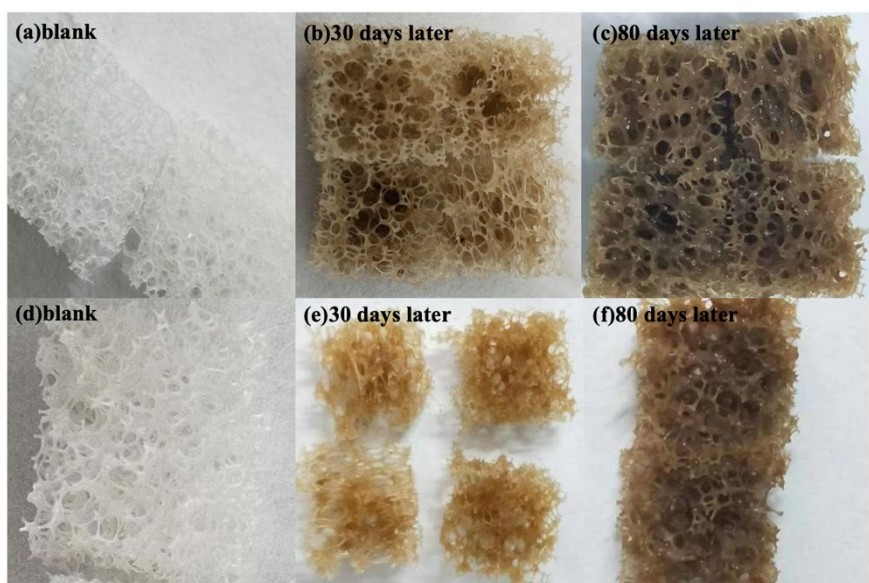

**Figure 5.** Change of the PBG carriers ((**a**–**c**) are complete and (**d**–**f**) are cut in half).

The results are shown in Figure 5d–f. The PBG carriers were cut and the microorganisms inside were observed and the microporous characteristics gave them excellent material diffusion and mass transfer efficiency. Combined with the treatment effect of the AMC–PBG/MBBR biological treatment process, the gradual enrichment of microorganisms on the carrier is the key to the effective removal of pollutants from wastewater by this process, indicating that the PBG cube has a good migration and diffusion ability for DO and organic matter, which ensures the stable growth and metabolism of microorganisms inside the sludge.

To further observe the growth of microorganisms in the carriers, the morphology and community changes of the sludge microorganisms were observed using microscopy. The observation results are shown in Figure 6.

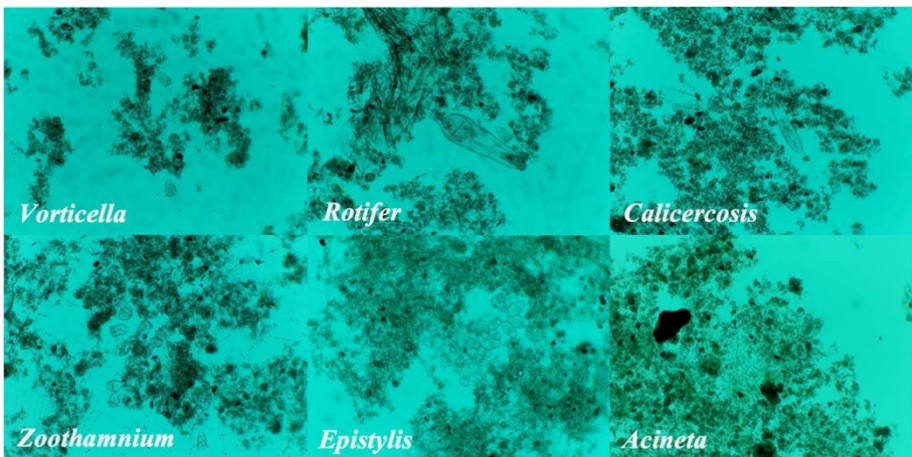

**Figure 6.** Microscopic examination of protozoan and metazoan animals.

During the stabilization period of the process operation, as shown in Figure 6, the protozoa and metazoan were mainly fixed ciliates, such as *Vorticella*, *Zoothamnium*, *Epistylis*, and *Acineta* as well as metazoans, such as *Rotifers* and *Nematodes*. Currently, the system has good water quality with low effluent COD and $NH_4^+$-N and high quality and mature activated sludge. In addition, higher organisms such as nematodes were present in the system, indicating that the system has some sludge reduction effect. Microfauna are microorganisms with higher nutrient levels compared to lower microorganisms. When energy is transferred from lower to higher microorganisms through predation, energy loss occurs, resulting in sludge reduction [34,35]. Therefore, the additional energy loss due to predation is the root cause of sludge reduction.

*3.4. Analysis of Microbial Diversity and Functional Microorganisms*

3.4.1. The Diversity

To describe the surface and internal structure of the carriers along with the microbial load in more detail, the two carriers were observed at different stages using SEM. The results are shown in Figure 7.

It can be seen from Figure 7 that the surface of the blank AMC (Figure 7a) has obvious concave and convex grooves and a fine pore-like structure which is conducive to the growth of microorganisms. According to the cross-sectional view, there are many pores inside which help microorganisms to grow inside and increase the sludge retention rate in the MBBR process. After 80 days, a large number of microorganisms attached to the surface of the carrier ran (Figure 7c), namely mainly filamentous and spherical bacteria, and a large number of protozoan whirling bacteria were observed. Figure 8 shows that the small number of microorganisms attached to the AMC may be due to the fact that the larger surface area of the AMC has been sufficient for the growth of microorganisms in the AMC–MBBR reactor and microorganisms did not need to overcome micropore diffusion to grow and metabolize within the AMC particles.

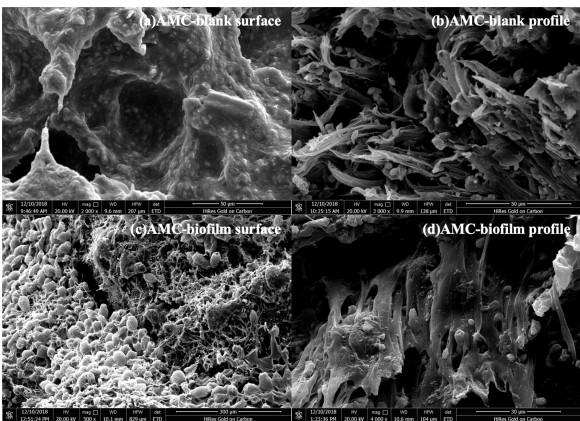

**Figure 7.** SEM image of AMC microorganism enrichment.

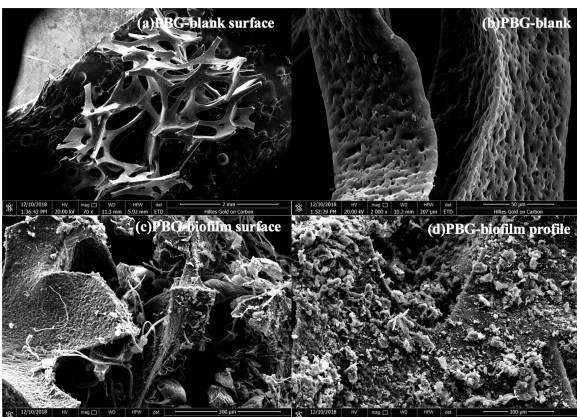

**Figure 8.** SEM image of PBG microorganism enrichment.

As seen in Figure 8, the blank PBG (Figure 8a,b) is a through-wall skeletal structure with microporous pores covering the surface of the skeletal wall, which facilitates the adsorption and growth of filamentous bacteria. After 80 days of operation, many microorganisms such as cocci and filamentous bacteria attached to the surface and the inside of the PBG. This indicated that various microorganisms gradually adhered from the surface to the interior to reach high concentrations of bacteria, which facilitated the efficient degradation of contaminants.

In summary, after a period of operation, the sludge microorganisms treating domestic wastewater gradually adhere and grow on the surface and inside of the two new carriers AMC and PBG. Organic matter and oxygen reach the internal space by diffusion to achieve stable growth and the metabolism of nine internal microorganisms. The application of the two carriers increases the amount of sludge in the system and may increase the microbial diversity, which facilitates the treatment of domestic wastewater with a composite biological process. At the end of the operation, the sludge capacity of the AMC and PBG reactors was calculated to be 0.427 g/g (based on VSS per gram of AMC) and 7.92 g/L (based on VSS per liter of PBG), respectively, indicating that both carriers have a good ability to enrich the microorganisms in the system.

### 3.4.2. The Classification

To understand the diversity of bacteria, archaea, and eubacteria during the MBBR bioprocess, an OTU-based alpha diversity analysis was performed based on the Shannon and Simpson indices (Table 3). The diversity indices clearly revealed changes in the composition of bacterial, archaeal, and fungal communities during digestion. The Shannon and Simpson diversity indices indicated a relative decrease in archaea, bacteria, and

eubacteria diversity during the steady-state operation. In other words, the system was screened by discharging suspended sludge to retain the dominant strains.

**Table 3.** Seq num, Shannon, and Simpson diversity index results for each sample.

| Name | Samples | Seq Num | Shannon | Simpson |
|---|---|---|---|---|
| Archaea | Activated sludge | 60,151 | 3.31 | 0.09 |
| | Sludge on carrier | 52,658 | 2.75 | 0.16 |
| | Suspended sludge | 42,749 | 3.29 | 0.09 |
| Bacteria | Activated sludge | 57,189 | 5.23 | 0.03 |
| | Sludge on carrier | 78,116 | 4.27 | 0.08 |
| | Suspended sludge | 49,601 | 3.86 | 0.13 |
| Eubacteria | Activated sludge | 72,319 | 2.55 | 0.15 |
| | Sludge on carrier | 72,078 | 1.89 | 0.30 |
| | Suspended sludge | 44,510 | 2.47 | 0.17 |

(1)   *Archaea*

The phylum *Euryarchaeota* (78–91%) was found to be the dominant taxon in all samples of the pale boundary (Table 4). In the MBBR system, the most abundant genera included *Methathrix* (40–56%) and *Methanegula* (11–16%) (Figure 9a). $H_2/CO_2$ and formate are available for their growth and methane production [36] and contribute to the degradation of pollutants in the effluent.

(2)   *Bacteria*

The most abundant phyla in the MBBR system were *Proteobacteria*, *Bacteroidetes,* and *Firmicutes* (Table 4), where the dominant species at the genus level were *Pseudomonas* and *Acinetobacter* (Figure 9b). *Bacteroidetes*, *Firmicutes*, and *Proteobacteria* were reported to be among the most common phylum and are mainly involved in the fermentation of complex polysaccharides [37].

(3)   *Eubacteria*

The most dominant phylum was unclassified, with 74.62% in the case of activated sludge, 47.24% in the case of sludge on the carrier, and 59.81% in the case of suspended sludge among them. Additionally, they were followed by *Nematoda* in activated sludge (16.45%), which was replaced by *Cryptomonadales* in sludge on the carrier (42.43%) and suspended sludge (17.36%) (Table 4). At the genus level, the distribution is basically similar: about half of the eubacteria are unclassified. In activated sludge, the remaining part mainly contains *Epistylis* (31.08%) and *Semitobrilus* (16.34%). In sludge on the carrier, the remaining part mainly contains *Cryptomonas* (42.54%) and *Cryptosporidium* (5.12%). In suspended sludge, the remaining part mainly contains *Cryptomonas* (18%) and *Semitobrilus* (9.97%) (Figure 9c).

In archaea, bacteria, and eubacteria, there are numerous and diverse microorganisms clustered within the biocarriers. All these contribute to denitrification and biodegradation of organic compounds [31].

**Table 4.** Comparative analysis of the three samples at the phylum level.

| Name | Phylum | Activated Sludge (%) | Sludge on Carrier (%) | Suspended Sludge (%) |
|---|---|---|---|---|
| Archaea | *Euryarchaeota* | 78.46 | 90.73 | 85.06 |
| | *Woesearchaeota* | 9.02 | 2.45 | 6.87 |
| | *Crenarchaeota* | 6.9 | 1.88 | 4.2 |
| | *Pacearchaeota* | 5.3 | 0.48 | 1.49 |
| | *Verrucomicrobia* | 0.02 | 1.64 | 1.12 |
| | *Thaumarchaeota* | 0.09 | 1.8 | 0.09 |

**Table 4.** *Cont.*

| Name | Phylum | Activated Sludge (%) | Sludge on Carrier (%) | Suspended Sludge (%) |
|---|---|---|---|---|
| | *Proteobacteria* | 53.06 | 80.46 | 75.32 |
| | *Bacteroidetes* | 12.01 | 8 | 15.92 |
| Bacteria | *Firmicutes* | 10.95 | 2.03 | 3.29 |
| | *Planctomycetes* | 3.08 | 2.32 | 1.06 |
| | *Nitrospirae* | 1.35 | 2.3 | 0.99 |
| | *Actinobacteria* | 1.79 | 1.81 | 0.35 |
| | *unclassified* | 74.62 | 47.24 | 59.81 |
| | *Cryptomonadales* | 2.6 | 42.43 | 17.36 |
| Eubacteria | *Nematoda* | 16.45 | 1.51 | 15.86 |
| | *Apicomplexa* | 0.16 | 5.14 | 4.67 |
| | *Ascomycota* | 1.83 | 1.67 | 1.34 |
| | *Mucoromycota* | 2.39 | 0.77 | 0.54 |

*3.5. Sludge Reduction Effect*

The excess sludge yield (Y) was calculated using Equation [38]:

$$Y = \frac{g\ SS_{end} - g\ SS_{start}}{g\ COD_{removed}} \tag{1}$$

The variable g SSstart indicates the total mass of SS present at the beginning of the test and g SSend indicates the sum of the mass of SS present at the end of the test plus the mass of SS from the wastewater and waste sludge leaving the reactor. After monitoring the COD removal and sludge discharge of the AMC–PBG/MBBR biological treatment process during steady operation, the apparent yield of process sludge was 0.043 kg MLSS/kgCOD.

The apparent yield of porous carriers and conventional activated sludge is about 0.3–0.4 kgMLSS/kgCOD [39], which has a significant impact on sludge reduction. This was mainly due to the fact that the multistage MBBR process provides an aerobic environment with the formation of different biological communities in the reactor and the complex biological communities lead to the reduction of energy units and residual sludge in the process. At the same time, the carriers form aerobic external and anoxic internal. In the oxygen microenvironment, organic pollutants were adsorbed on the carrier surface. The surface microorganisms consume nutrients and energy through biological oxidation reactions, decomposing dissolved organic matter in the wastewater, and the decomposition and oligomerization of sludge will release carbon sources, promoting denitrification effects and achieving in situ sludge reduction [40]. Secondly, for general heterotrophic microorganisms, their growth mode is usually "direct utilization", that is, in an aerobic environment with sufficient organic substrates. Due to the dominant flora in the AMC–PBG/MBBR biological treatment process, which is an aerobic and anoxic mode, common heterotrophic bacteria are at a disadvantage and grow slowly [41]. In this way, the sludge reduction is achieved.

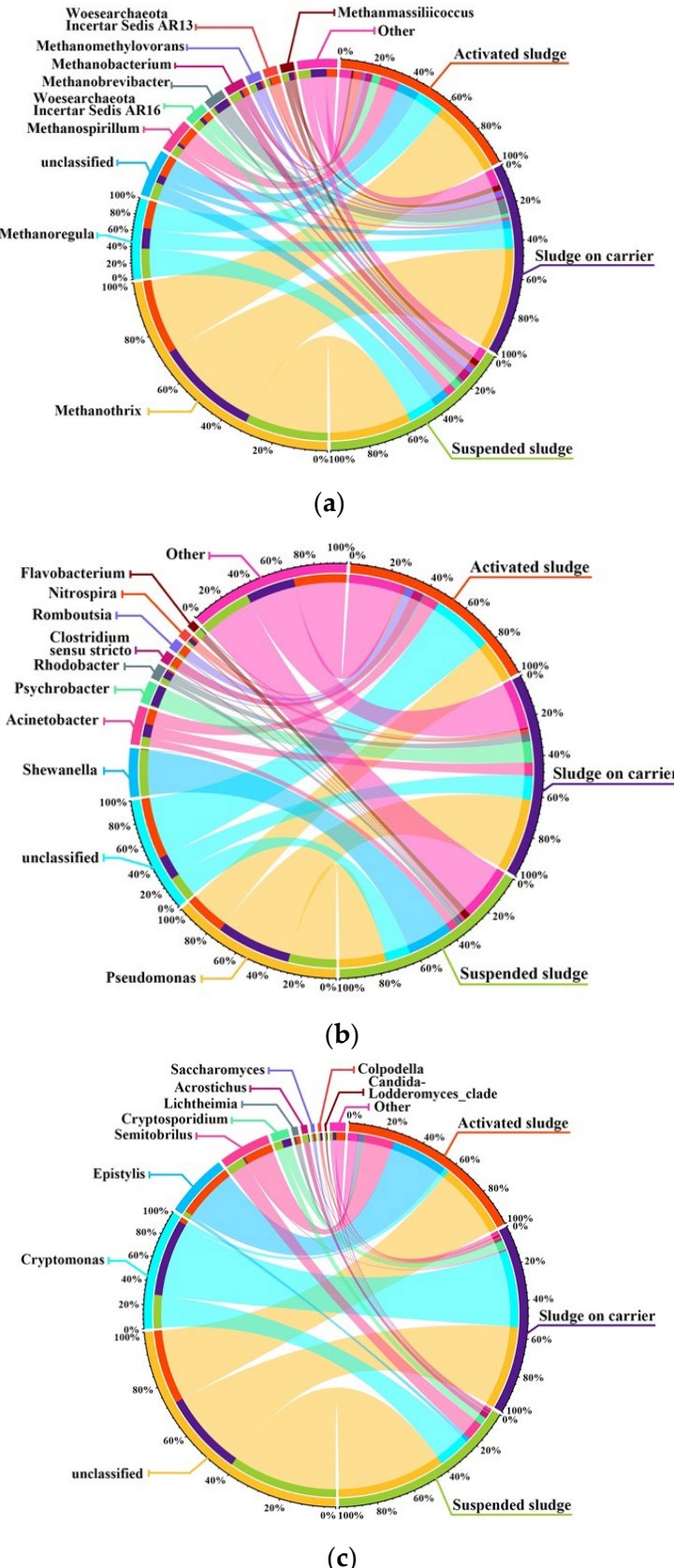

**Figure 9.** Microbial community diversity at phylum level for (**a**) archaea, (**b**) bacteria, and (**c**) eubacteria.

## 4. Conclusions

Two different types of biocarriers of composite multi-stage MBBR were successfully established, both of which could complete the rapid start-up process in a short time with final effluent concentrations of 44.2 mg/L, 0.81 mg/L, and 10 mg/L for COD, $NH_4^+$-N and $BOD_5$, respectively. Upon maintenance of a high OLR (1.5 kgCOD/(m$^3$·d)) and room temperature, the two types of biocarriers had strong microbial enrichment capacity, with various archaea, bacteria, and eubacteria degrading COD and $NH_4^+$-N in domestic wastewater, such as *Euryarchaeota*, *Proteobacteria,* and *Cryptomonadales* mainly present in the filler to form biofilms. The AMC–PBG/MBBR process is a high-efficiency and low energy-consumption treatment of domestic wastewater because it can simultaneously remove nitrogen, phosphorus, and carbon. Moreover, there was also a reduction in sludge, with an apparent sludge yield of only 0.043 kgMlss/kgCOD and a 91.4% reduction compared to conventional activated sludge. To further verify the applicability of the process, pilot scale studies are needed to provide valuable information for parameter optimization and improved efficiency and automation before applying the AMC–PBG/MBBR process to treat actual domestic wastewater for a long time. In addition, attention should be paid to the possibility of using the AMC–PBG/MBBR process to treat other types of wastewaters.

**Author Contributions:** Conceptualization, M.L.; Methodology, Y.L. and X.Z.; Resources, N.W.; Data curation, N.W.; Writing—original draft, M.L.; Funding acquisition, Y.L., N.W. and B.Y. All authors have read and agreed to the published version of the manuscript.

**Funding:** This research was funded by the Science and Technology Plan Projects of the Shannxi Provincial Water Resources Department (Grant No. 2017 slkj-9), the National Natural Science Foundation of China (Grant No. 22008188 and No. 42107493).

**Data Availability Statement:** Not applicable.

**Acknowledgments:** The authors are grateful to the anonymous referees who provided valuable comments and suggestions to significantly improve the quality of the paper. The authors would like to extend sincere appreciation to the support by the Science and Technology Plan Projects of the Shannxi Provincial Water Resources Department (Grant No. 2017 slkj-9), the National Natural Science Foundation of China (Grant No. 22008188 and No. 42107493).

**Conflicts of Interest:** On behalf of all authors, the corresponding author states that there is no conflict of interest.

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
