# Peer review of "A Study on The Carriers Compound Multi-Stage MBBR Biological Treatment Process for Domestic Sewage"

_sustainability, doi:10.3390/su15107922_

Round 1

Reviewer 1 Report

The manuscript entitled “Study on Carriers Compound Multi-stage MBBR Biological Treatment Process for Domestic Sewage” has studied the degradation process of actual domestic wastewater by MBBR treatment process with two carriers, composite AMC and PBG. Here with some specific comments. 

1.     Abstract: is not certain and accurate concerning the contents of the article, add some information, such as methods.

2.     Line 19: What does “adsorption capacity” mean here? Better give a more accurate description.

3.     Line 20-21: It is necessary to give the concrete data of sludge reduction assessment.

4.     Line 11: All abbreviations should be interpreted when they first appear.

5.     Introduction: it is suggested to summarize and condense the section on the status of the study.

6.     Line 78, 83, 87: another problem with the abbreviation of proper noun formatting needs to be revised.

7.     Section 2.1: it is suggested that section 2.1 be organized and supplemented or combined with later sections.

8.     Line 136, 144: Use the same way for referring to things, for example, 5 days and not 5 d.

9.     Figure 2: why did the NH4+-N removal rate suddenly drop on days 17, 18 and 24? Need explanation.

10.  Line 352-351: please supplement and analyze the cause of formation.

11.  Figure 9: Text in Figure 9 is not legible, please improve the resolution of Figure 9.

12.  Conclusion: The summary in the conclusion section is not good enough, it is recommended to make modifications

13.  This manuscript has some problems with expression and grammar. The language needs to be improved to meet publication standards.

Reviewer 2 Report

As a result of the growth and development of the urban population, the volume of domestic and industrial wastewater is increasing. In this regard, the amount of waste generated, including excess activated sludge, from the biological wastewater treatment plant is also growing. The resulting excess activated sludge accumulates on silt maps, which occupy large areas. The stored sediment is infected with dangerous bacteria that can cause various forms of infectious diseases, contains a large number of helminth eggs, heavy metal compounds of various shapes. The very operation of silt maps leads to the loss of valuable lands, soil pollution, the spread of unpleasant odors, the accumulation of heavy metal salts, as well as the spread of negative microbiological and gas background, which negatively affects the environment and human health.

In this regard, methods and technologies that reduce the formation of sediment and their own excess activated sludge require special attention, and the presented publication is relevant.

The article can be recommended for publication.

Remarks.

1. Additions are required at the place of taking the material for analysis and analysis of wastewater data. «Domestic sewage and activated sludge were collected from a drainpipe at Xi’an Polytechnic University (Xi’an, China).» 

2.Conclusion "Abundant microbial populations were attached to the carriers to reduce sludgeproduction. The apparent sludge yield was only 0.043 kg Mlss/kg COD, which was a sig- nificant sludge reduction compared with the conventional activated sludge method." It is necessary to specify in detail how much the amount of sludge is reduced compared to the traditional method of obtaining activated sludge.

Author Response

We greatly appreciate for your careful review and earnestly constructive suggestions regarding our manuscript “Study on Carriers Compound Multi-stage MBBR Biological Treatment Process for Domestic Sewage”. These comments are all valuable and helpful for improving our manuscript, and have an important guiding significance to our research work. We have studied comments carefully and tried our best to revise and improve the manuscript. The revised portion is marked in blue in the revised manuscript. The point-to-point answers and explanations for all comments are listed in the following. We hope that these corrections will meet with the approval. These changes will not influence the content and framework of the paper and we hope that the correction will meet with approval.

We appreciate for Editors and Reviewers’ warm work earnestly and hope that the correction will meet with your approval. Once again, thank you very much for your comments and suggestions and look forward to hearing from you soon.

Yours sincerely,

Miaojie Li, Yonghong Liu, Xiaode Zhou, Ning Wang, Bo Yuan.

April 28, 2023

Response to Reviewers #2

Point 1: Additions are required at the place of taking the material for analysis and analysis of wastewater data. «Domestic sewage and activated sludge were collected from a drainpipe at Xi’an Polytechnic University (Xi’an, China).»

Response 1: Thanks for your suggestion. We have added. Specifically reflected in Line 88-91. The specific modifications are as follows:

Domestic sewage and activated sludge were collected from a drainpipe at Xi’an Polytechnic University (Xi’an, China). The basic data of domestic sewage are listed in Table 1.

Point 2:.Conclusion "Abundant microbial populations were attached to the carriers to reduce sludge production. The apparent sludge yield was only 0.043 kg Mlss/kg COD, which was a significant sludge reduction compared with the conventional activated sludge method." It is necessary to specify in detail how much the amount of sludge is reduced compared to the traditional method of obtaining activated sludge. 

Response 2: Thanks for your suggestion. We have added the calculation of sludge reduction versus the traditional method in the conclusion section. Specifically reflected in Line 379-381. The specific modifications are as follows:

The surface microorganisms consume nutrients and energy through biological oxidation reactions, decomposing dissolved organic matter in the wastewater, and the decomposition and oligomerization of sludge will release carbon sources, promoting denitrification effects and achieving in situ sludge reduction